# Beyond active learning: Using 3-Dimensional learning to create scientifically authentic, student-centered classrooms

**Melanie M. Cooper**[1]*, **Marcos D. Caballero**[2,3,4], **Justin H. Carmel**[5], **Erin M. Duffy**[6], **Diane Ebert-May**[7], **Cori L. Fata-Hartley**[8], **Deborah G. Herrington**[9], **James T. Laverty**[10], **Paul C. Nelson**[8], **Lynmarie A. Posey**[1], **Jon R. Stoltzfus**[11], **Ryan L. Stowe**[12], **Ryan D. Sweeder**[13], **Stuart Tessmer**[2], **Sonia M. Underwood**[5]

1 Department of Chemistry, Michigan State University, East Lansing, Michigan, United States of America, 2 Department of Physics & Astronomy, Michigan State University, East Lansing, Michigan, United States of America, 3 Department of Computational Science, Mathematics and Engineering, Michigan State University, East Lansing, Michigan, United States of America, 4 Department of Physics and Center for Computing in Science Education, University of Oslo, Oslo, Norway, 5 Department of Chemistry & Biochemistry and STEM Transformation Institute, Florida International University, Miami, Florida, United States of America, 6 Science Department, Solebury School, New Hope, Pennsylvania, United States of America, 7 Department of Plant Biology, Michigan State University, East Lansing, Michigan, United States of America, 8 Human Biology Program, Michigan State University, East Lansing, Michigan, United States of America, 9 Department of Chemistry, Grand Valley State University, Allendale, Michigan, United States of America, 10 Department of Physics, Kansas State University, Manhattan, Kansas, United States of America, 11 Department of Biochemistry and Molecular Biology, East Lansing, Michigan, United States of America, 12 Department of Chemistry, University of Wisconsin–Madison, Madison, Wisconsin, United States of America, 13 Lyman Briggs College, Michigan State University, East Lansing, Michigan, United States of America

* mmc@msu.edu

**Data Availability Statement:** All relevant data are within the manuscript and its Supporting Information files.

## Abstract

In recent years, much of the emphasis for transformation of introductory STEM courses has focused on "active learning", and while this approach has been shown to produce more equitable outcomes for students, the construct of "active learning" is somewhat ill-defined and is often used as a "catch-all" that can encompass a wide range of pedagogical techniques. Here we present an alternative approach for how to think about the transformation of STEM courses that focuses instead on what students should know and what they can do with that knowledge. This approach, known as three-dimensional learning (3DL), emerged from the National Academy's "A Framework for K-12 Science Education", which describes a vision for science education that centers the role of constructing productive causal accounts for phenomena. Over the past 10 years, we have collected data from introductory biology, chemistry, and physics courses to assess the impact of such a transformation on higher education courses. Here we report on an analysis of video data of class sessions that allows us to characterize these sessions as active, 3D, neither, or both 3D and active. We find that 3D classes are likely to also involve student engagement (i.e. be active), but the reverse is not necessarily true. That is, focusing on transformations involving 3DL also tends to increase student engagement, whereas focusing solely on student engagement might result in courses where students are engaged in activities that do not involve meaningful engagement with core ideas of the discipline.

**Funding:** Our funding acknowledgement National Science Foundation [EHR DUE 1725520], should be changed to National Science Foundation [Education and Human Resources, Division of Undergraduate Education 1725520]. The funders had no role in study design, data collection and analysis, decision to publish, or preparation of the manuscript.

**Competing interests:** The authors have declared that no competing interests exist.

## Introduction

Over the past twenty years, the wide array of pedagogical techniques that have come to be collectively known as active learning [1,2] have been the predominant focus for transformation efforts in higher education STEM teaching and learning. A meta-analysis across a wide range of STEM disciplines found that the use of active learning techniques tends to increase average course grades, particularly by decreasing the DFW (D's, F's and withdrawals) rate [3]. In addition, active learning appears particularly helpful in promoting more equitable outcomes for students [4,5]. Thus, many faculty development efforts focus on instructional practices that increase student engagement in the classroom [6–8], although studies have also shown that their uptake is somewhat disappointing [9].

However, as discussed in "The Curious Construct of Active Learning", an extensive overview and synthesis of the literature on active learning across STEM disciplines [1], the construct of active learning is amorphous; it can refer to minor adaptations to lecture-based courses, such as the use of student response systems, or to flipped classrooms, or to completely re-envisioned curricula taught in studio classrooms. It has become the catch-all description for a wide range of instructional practices. Freeman et al., in their highly cited meta-analysis on the impact of active learning [3] do not disaggregate findings by instructional strategy and use the following definition to determine which courses engaged students in active learning: "*Active learning engages students in the process of learning through activities and/or discussion in class, as opposed to passively listening to an expert. It emphasizes higher-order thinking and often involves group work.*" It is notable that these authors do not say more about *what* students are learning, other than a reference to higher-order thinking or critical thinking; constructs that are also subject to multiple definitions [10–12]. Thus, the often ill-defined construct of active learning is characterized by other ill-defined constructs. For example, Bailin noted that "the field lacks a coherent and defensible conception of critical thinking,"[13]. As noted in the National Academies report on 21st Century thinking skills, there is considerable evidence that "higher order thinking" or "critical thinking skills" are domain specific, rather than domain general–that is teaching such skills should be connected to domain knowledge [14].

Certainly, the evidence is clear that student engagement is a necessary, but perhaps not sufficient, component of learning for most students [15]. Evidence for the use of active learning strategies typically does not include discussions of *what is being learned* during the instruction, or how students are able to use that knowledge. The most highly cited study on the impact of active learning on student outcomes relies on scores on conceptual multiple-choice exams and/or course grades as evidence of improvement [3]. While there are certainly studies where what students must know and do are carefully described and measured, for example in reports of the impact of transformation projects such as Process Oriented Guided Inquiry Learning (POGIL) [16–18], or those that incorporate simulations that allow students to explore phenomena [19,20], it appears that active learning has often been conflated with having students engage with some classroom activity, regardless of what that activity involves. In many cases, little information is provided about what students are doing or what the subsequent assessments are based on [5]. We have previously argued that it is not sufficient to know facts (or even concepts) or to solve rote mathematical exercises; students should be able to use their knowledge to explain, model, and predict phenomena by engaging in Three-Dimensional Learning (3DL) [21].

3DL was originally developed in the National Academies consensus report "A Framework for K-12 Science Education" which lays out a vision for science education that centers the role of constructing productive causal accounts for phenomena [22]. The "Framework" proposes that three interconnected dimensions of science learning are central to the vision: Scientific and Engineering Practices, Disciplinary Core Ideas, and Crosscutting Concepts, as shown in Fig 1.

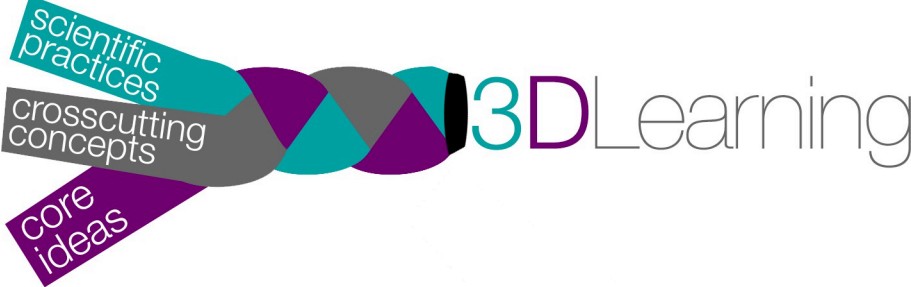

**Fig 1. 3D-Learning.** The three dimensions, Scientific and Engineering Practices, Disciplinary Core Ideas, and Crosscutting Concepts, are intertwined to form 3D-Learning.

The Scientific and Engineering Practices are the ways that scientists use and engage with their knowledge–for example, asking questions or defining problems, developing and using models, and evaluating and communicating information. The Core Ideas of a discipline are the underlying ideas that have broad applicability and can be used to predict and explain phenomena at different levels of depth and complexity. For example, in biology the core idea of evolution underlies a vast range of phenomena; atomic and molecular interactions and bonding play a similar role in chemistry, as do fields as the mediators of interactions in physics. The Crosscutting Concepts can be thought of as lenses that transcend disciplines and allow scientists to focus their investigation of a phenomenon on specific aspects–for example, cause and effect, structure and function, and stability and change [23]. The Framework emphasizes that, for meaningful learning, the three dimensions must be inextricably intertwined. For example, if we want our students to be able to model (a Scientific Practice) how energy (a Core Idea) is transferred within or between systems (a Crosscutting Concept), all three dimensions must be combined during instructional and assessment activities. A focus on content without considering how students should use that knowledge can lead to fragmented, disconnected understanding and inert knowledge that is not useful in new contexts [22]. Consequently, it is important to intentionally build all three dimensions into learning objectives, assessments, and classroom activities.

There are several other approaches to transformation of STEM education that, while not explicitly developed using a 3D framework, also have the potential to engage students in the use of scientific practices. For example, POGIL activities often begin with experimental data that students may analyze and interpret to make an argument about the phenomenon under study [17]. Similarly, Course Based Undergraduate Research Experiences (CURES), may ask students to design and carry out experiments, analyze and interpret data and so on [24,25]. 3DL is a unifying framework that can tie all these approaches together.

The 3DL approach has been the focus of a multidisciplinary project to transform the STEM gateway (introductory biology, chemistry and physics) courses at Michigan State University (MSU) by adapting the Framework for use in higher education [21]. In our work, we adopted the Scientific Practices and Crosscutting Concepts as presented in the Framework with minor modifications. However, the Core Ideas presented in the Framework were originally developed for a more interdisciplinary approach to K-12 science teaching and learning (for example, Core Ideas are presented for physical science), which does not align with how introductory science courses are organized at the college level. Furthermore, the Core Ideas were not intended for the depth of college science courses. We worked collaboratively with disciplinary experts and introductory course instructors from chemistry, biology, and physics to define more appropriate Core Ideas for those disciplines [26,27].

To evaluate the extent of the transformation, we developed two protocols, the Three-Dimensional Learning Assessment Protocol (3D-LAP) [27] and the Three-Dimensional Learning Observation Protocol (3D-LOP) [28], that are intended to characterize the extent to which assessments and instruction incorporate 3D learning. The 3D-LAP allows us to determine whether an assessment task has the potential to elicit a 3D response from students. We have previously reported on the use of the 3D-LAP as a tool to evaluate the change in 3D assessment over time [29] by coding over 4,000 examination questions from midterm and final exams that were used in the introductory courses. These earlier findings show that the 3D-LAP is a useful tool for characterizing the extent of transformation which focuses on student use of knowledge in the context of Core Ideas. Our findings showed that for a large general chemistry course that was completely transformed using the 3DL approach [30], with around 50% of exam points associated with 3D tasks on course exams, the average grades in the course increased, and the percentage of students who received a D, F or W in the course decreased [29]. That is, we saw similar overall outcomes to those reported for courses that employ active learning. Another study showed that 3DL assessment tasks focusing on mechanistic reasoning about a chemical phenomenon are more equitable than typical general chemistry tasks involving calculations that are more traditionally featured on such exams [31].

We note that the 3D-LAP can only give us a measure of *the potential* of an assessment item to elicit 3DL. To determine whether students are actually engaging in 3DL requires that we analyze student responses to such assessment tasks, and study how students construct these responses. There are now a number of studies in which responses from matched cohorts of students, from both traditional and transformed sections of a course, were analyzed [32–37]. In general, these studies show that students in 3DL courses are significantly more likely to engage in construction of mechanistic explanations and construction and use of models in the context of 3D tasks. Another study asked students "what kind of thinking" they were expected to engage in for a given course, and what kinds of thinking course assessments tested [38]. Students in a 3DL-transformed organic chemistry course were most likely to respond that they were expected to use their knowledge (apply and reason), whereas students in a traditional section were more likely to perceive that they were expected to memorize information. In a study that followed students through four semesters of chemistry, (the first two semester were 3D and the last two were traditional) we found that students from the very large enrollment 3D general chemistry course performed as well or better than students from more selective courses (honors or majors) in the next two organic courses in the sequence [32].

One intriguing study compared three student cohorts from different institutions on a task that asked students to explain how and why a substance dissolved in water [39]. All instructors agreed that students would have learned about this phenomenon and should be able to complete the task. The three different institutions employed three different instructional approaches: 1) a traditional lecture (didactic) course with a traditional curriculum, 2) an active learning approach with a traditional curriculum, and 3) a 3DL approach with transformed curriculum and assessments that included 3D tasks. Students in the didactic and active learning courses provided similar responses, while the 3DL students were far more likely to construct a full explanation for the phenomenon, invoking ideas about both interactions and bonding, and energy changes. Although we should not be surprised that students who were enrolled in traditional courses were less able to provide appropriate responses to a 3DL task (students tend to learn what is emphasized in a course), we also note that the active learning students performed similarly to those who listened to lectures. This indicates that active learning alone does not support students' understanding of mechanistic reasoning and chemical phenomena if the curriculum does not intentionally include activities that require students to use knowledge in this way.

3DL can also support the integration of important and overlooked scientific practices. Using a computer to model scientific phenomenon and data is central to the enterprise of science, and yet, computational modeling is absent from most introductory science instruction. Some introductory physics courses use computational modeling activities to support students as they make predictions or construct explanations [40]. Instructional strategies used in computer science education, such as pair programming [41,42] and live-coding [31], have been used in disciplinary courses to ensure students are active in their learning of computational modeling. Research conducted in our introductory physics courses has demonstrated that such active learning strategies alone are not sufficient to promote 3DL. However, newer computationally enabled physics courses at MSU that are characterized as 3D and are designed around a communities of practice framework [32,33] have been shown to support the ways in which students work in their groups as they develop computational models for real-world phenomena [42–47].

While analyzing the types of assessment tasks and concomitant student responses has provided us with compelling evidence for the efficacy of 3DL, it does not provide information about the instructional practices employed during instructional activities. To address this need, we developed the 3D-LOP, which can be used to characterize instruction by coding classroom video recordings. The 3D-LOP is, in some ways, similar to other classroom observation protocols, such as the COPUS or RTOP [48,49], because it provides a way to characterize the instructional methods used in a classroom. What sets the 3D-LOP apart from other such observation protocols is that it also allows the user to characterize *what is being taught* as well as *the way that it is taught*. Thus, the 3D-LOP provides a way to characterize a class period by the topic covered, instructional activities, whether the instruction is aligned with 3DL, and whether that instruction is student- or instructor-centered. Full details of the development and coding protocols for the 3D-LOP have been published previously [28].

Using class video recordings, we are now able to investigate the enactment of 3DL in these introductory biology, chemistry, and physics classes and to determine whether there is a connection between 3DL and the more common approach to reform that typically focuses on instructional practices and student engagement (i.e., active learning). We have evidence that transformation efforts using different approaches (e.g., 3DL or active learning) do tend to result in some of the same outcomes (for example, improved average grades, retention rates, and more equitable outcomes). However, as noted earlier, there are aspects of 3DL that go beyond active learning. For example, students who are enrolled in courses where active engagement, but not where 3DL is prevalent, are unlikely to provide causal explanations for phenomena [39]. Additionally, 3DL supports inclusion of scientific practices that are often neglected, such as computational modelling [40]. It also explicitly defines what is expected as outcomes for a course; rather than "knowing" or "understanding," students perceive that they are going to apply and reason with their knowledge [38].

Now that we have the methodology to characterize whether 3DL and/or active learning take place during class instruction, the current study uses evidence from class session video recordings to address the following research questions:

1. In what ways can 3DL and/or active learning occur within a STEM course?

2. How are 3DL and active learning related?

## Experimental methods

The data corpus explored in this report is composed of video recordings of in-class instruction collected over 4 sequential years from introductory biology, chemistry, and physics courses at

Michigan State University. As discussed in our earlier paper [28] the camera was mounted at the back of the classroom to record the instructor activities and interactions between instructor and students. All the instructors in our data set gave permission for their classes to be video recorded and for their course exams to be analyzed. All identifying information, such as instructor name, course ID, semester, etc. were removed before coding, and randomly generated identifiers were applied to the video files, in accordance with the Michigan State University approved IRB protocols (IRB# x13-1110e Category: Exempt 1, IRB# x17-1053e Category: Exempt 1).

### 3D-LOP analysis of classroom videos

The 3D-LOP was used to code video recordings, as discussed in our prior work [28]. For the present study, videos of 82 class meeting sessions across the gateway courses in biology, chemistry, and physics were recorded. Each video was segmented into blocks of contiguous time devoted to a particular topic of instruction. These segments (N = 417) were then coded by two researchers from the team for alignment with the three dimensions and whether instruction was instructor- or student-centered. By instructor-centered, we mean the instructional activities were lecture-based, perhaps with call-and-response questioning. By student-centered, we mean more extensive instructor-student interactions, group or individual tasks, clicker questions and so on. This process allowed us to create parallel sets of timelines that provide information about the topic being taught, the classroom activities, whether the instruction was 3D, and whether the topic segment was "student-centered" or "instructor-centered". Here we define "active" segments as those with more than 50% of the time dedicated to teaching activities that directly engaged students (interactions, tasks, clicker questions), and 3D segments as those that spend more than 50% of the time on 3DL. We chose these criteria for simplicity and because other researchers have also used this cut off to determine whether active learning is present [5]. Thus, a segment may be characterized as 3D and active, active only, 3D only, or neither 3D nor active. The data reported here do not include those used in our development of the 3D-LOP and were recorded over several years before the COVID pandemic.

Each video was coded by at least two disciplinary experts, and if the IRR fell below 70% a third disciplinary expert also coded the videos. Eventually all videos were coded until all the coders agreed.

## Results and discussion

### RQ 1: In what ways can 3DL and/or active learning occur within a STEM course?

As discussed above, the data from coding class videos with the 3D-LOP allow us to develop visualizations of class timelines as shown in Fig 2. Each timeline shows the types of instructional activities (teaching activities) and the segments of class (or topics) that are three dimensional. Each 3DL segment is coded as instructor-centered (l) or student-centered (S), which allows us to determine: 1) the class time that is devoted to student engagement (active learning), 2) the class time that is devoted to 3DL, and 3) the class time that is devoted to both 3DL and active learning. Fig 2 provides representative examples of such timelines, which can be characterized as (a) active but not 3D, (b) active and 3D, (c) 3D but not active, and (d) neither active nor 3D.

**Active but not 3D.** Instructional segments that employ active learning techniques at least 50% of the time but are not 3D are quite common in our data set (95 of 417 segments, see Fig 3); the timeline in Fig 2A exemplifies one such class session. In this introductory biology class

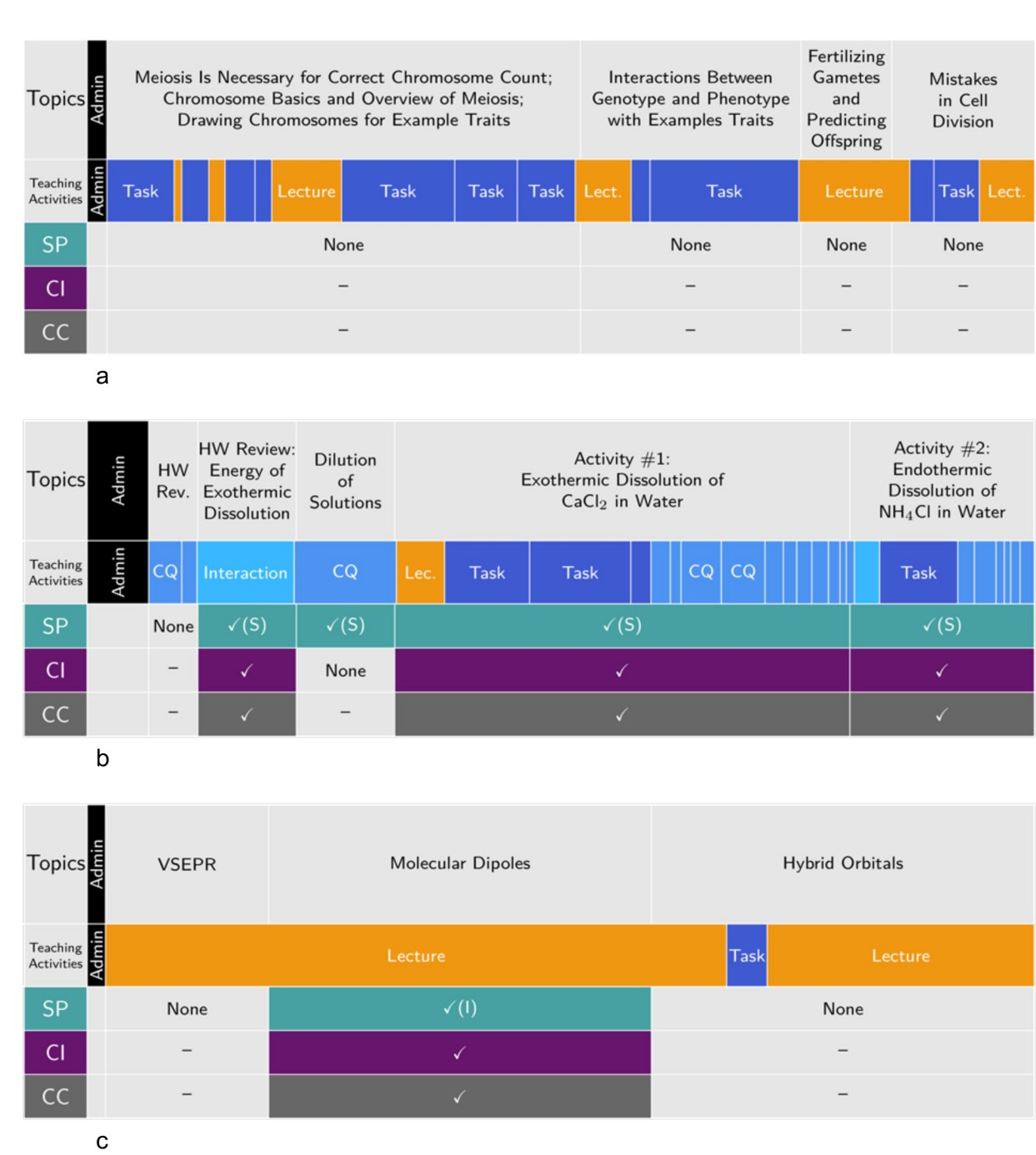

**Fig 2. Representative timelines highlighting examples of each of the four possible combinations of active and 3D class segments.** (a) A 73-minute biology class session that is active but not 3D. (b) A 77-min long chemistry class session that is both active and aligned with 3DL. (c) A 47-min long chemistry class session that is not active but contains a 3DL-aligned segment. (d) A 52-min physics class that is neither active nor 3D. The leftmost column provides a label for each row, with the beginning of class starting at the right edge of the label column and the end of the class session at the right edge of the timeline; the width of each segment (labelled here as Topic) is directly proportional to the time spent on that topic during the class session. Admin refers to announcements by instructor; Task refers to students engaging in a task as described in the 3D-LOP paper. Checkmarks and filled color denote the presence of a given dimension with "(I)" and "(S)" denoting Instructor or Student-centered engagement with a Scientific Practice, if present.

| | Not Active | Active | Total |
|---|---|---|---|
| **Not 3D** | 2.66<br>Expected: 205<br>Observed: 229 | -4.90<br>Expected: 119<br>Observed: 95 | 364 |
| **3D** | **-9.78**<br>**Expected: 59**<br>**Observed: 35** | **16.6**<br>**Expected: 34**<br>**Observed: 58** | 93 |
| **Total** | 264 | 153 | 417 |

Color key for standardized residual value

**Fig 3. Contingency table displaying the number of segments observed to be 3D or not, versus active or not.** A segment is defined as active when 50% or more of the segment time is devoted to non-lecture teaching activities, and as 3D when 50% of more or the segment time involves 3D instruction. Expected counts and the chi-square residuals for each cell are given above the observed counts. Cell color indicates the magnitude of the contribution of the residual to the chi-square value for the table as shown in the color bar below the table: Blue indicates more than expected by chance and red less than expected. The chi-square value for the table is 33.9, with the Fisher's p-value of less than 0.001 and an effect size (phi) of 0.285 (small to medium). The critical Bonferroni-corrected chi-square value is 9.14 for an alpha of o,01.

session, students worked in groups on several activities, which we label as "Tasks". These tasks included discussions on identifying the correct number of chromosomes for a given set of cells, drawing cells in different stages of cell division as the discussion of meiosis progressed, and identifying the genotype and phenotype of common traits (e.g., mid-digital hair). Punctuating these task segments were periods of lecturing in which the task is concluded or new topics are introduced. The tasks were focused on observation of elements of biological systems and building foundational skills. Although these tasks potentially build important foundational knowledge for biology and prepare students for future topics in the class, they do not engage students with any of the Scientific Practices. While most of the class period is devoted to student activities and is engaging students in a range of tasks, it is not asking students to use their knowledge in sensemaking activities, but rather the students are re-representing ideas in the form of pictures and diagrams.

**Active and 3D.** The data set contains 58 segments that are both active and 3D. An example of a class with 3 such segments is provided by the timeline in Fig 2B from a general chemistry course, in which the topic of solutions is discussed over several segments. Here, we see instructional approaches that included instructor-student interactions that go beyond lecture interactions, multiple tasks that engage students, and a series of clicker questions followed by group discussions. Aside from the segment designated as "Homework Review", all topics were explored by means of at least one of the Scientific Practices, and 3 out of 4 of these segments in the class are three-dimensional. All the occurrences of Scientific Practices are characterized as student-centered. Students constructed explanations and models for themselves, rather than watching the instructor work through the reasoning. For instance, the activities focused on students engaging with a hands-on activity (observing the dissolution of a salt and the accompanying temperature change). After scaffolded group discussions facilitated by the instructor, Graduate Teaching Assistants and Undergraduate Learning Assistants supported groups of

students working to create an explanation for their observations and answer a series of clicker questions. By incorporating the Core Ideas of *Energy* and *Electrostatic and Bonding Interactions* and the Crosscutting Concept of *Systems and System Models*, students constructed a model that explained the observed temperature change, by relating it to the energy changes as various interactions are either broken or formed. While not all class sessions that are 3DL-aligned are as active, the co-occurrence of 3DL and active learning appears to be common in 3D class sessions as will be discussed later.

**3D but not active.** Not all classes that focus on 3DL are also student-centered (35 of 417 segments), as illustrated by the timeline shown in Fig 2C, which depicts a traditional lecture in a large-enrollment general chemistry course. The class consisted of three different topic segments. Instruction began with a lecture review of Valence-Shell Electron-Pair Repulsion (VSEPR) Theory, followed by using that theory to understand how molecular dipoles can arise and how they can be used to predict molecular properties. Finally, hybrid orbital formation in bonding was described. After the VSEPR review, the instructor engaged students with the content by combining use of VSEPR as a model that predicts molecular shape and knowledge of bond polarities to predict the distribution of electron densities in a molecule. Molecular polarity was then used to predict and explain molecular properties. Although some student engagement occurred during this class session, it was either via call-and-response questioning or by the instructor addressing student questions during the lecture; therefore, we do not characterize the instruction in this class as active learning, since it did not incorporate activities usually designated as active learning, such as clicker questions, peer discussion, or group activities [1,3].

The non-active module on the use of molecular dipoles has been characterized as three-dimensional. However, in this case, the instructor was doing the work of demonstrating how to explore a phenomenon (why different molecular substances have different melting and boiling points), while employing the Scientific Practice of *Developing and Using Models* for their students (as indicated by the "(I)" on the diagram meaning an "instructor-centered segment"). By discussing the molecular shape and electron distribution, the instructor showed how the molecular structure can be used to predict molecular properties (a Core Idea), by using VSEPR as a model and the Crosscutting Concept of cause and effect (strong attractive interactions between polar molecules cause the macroscopic substance to exhibit predictable properties such as a high boiling point).

**Neither active nor 3D.** We have numerous examples of classroom recordings that are neither active, nor 3D (229 of 417 segments). Fig 2D provides an example of a physics class that meets this criterion. In this traditional-lecture, large-enrollment introductory electromagnetism course, the lecture focused on introducing the fundamental connections between electric fields and charges. These ideas are certainly important and foundational to much of the future curriculum. However, the instructor did not present the phenomena and key connections in a way that employed any Scientific Practices; hence, the class was not 3D. Although some student engagement occurred, it was either call-and-response questioning or the instructor addressed student questions during the lecture. Whereas many of the traditional-format physics courses in our database employed clicker questions and peer discussion, in this case the instructor did not engage students in this way. Hence, this class was not designated as including any active learning (Freeman et al., 2014; Lombardi et al., 2021).

The four examples of class timelines in Fig 2 show that active learning and 3DL are in fact different, and that these differences can be detected using the 3D-LOP. It is possible to have one without the other; a class session may be highly active and yet not involve students' use of knowledge to make sense of phenomena. For example, the class in which students draw the stages of meiosis (shown in Fig 2A) has students engaged in tasks most of the time, but they

are not engaged in reasoning about Core Ideas. Rather, they are drawing diagrams of a process, and those diagrams are not used to predict or explain anything further. On the other hand, Fig 2B represents a class which is both active and involves 3DL where students are engaged in sensemaking activities that are 3D. Students feel the temperature change when a salt dissolves in water and use the core ideas of interactions and energy to predict and explain this phenomenon. Fig 2C represents a class where there is a 3DL segment in which the instructor is doing the work by modelling for the class how VSEPR can be used to predict and explain molecular properties, but there is little meaningful input or activity from most of the students. Finally Fig 2D represents a class where there are no student-instructor interactions, the material is being delivered by a lecture, and does not include any 3D segments.

## RQ 2: How are 3DL and active learning related?

As shown in the examples above, active learning and 3DL are different, and it is entirely possible to have one without the other. An important next question, then, is: how are active learning and 3DL related? To address whether there is a relationship between active learning and 3DL in our data set, we designated the 417 video segments as active or not active, and 3D or not-3D, using the criterion that 50% of the instructional time in the segment should be active and/or 3D, respectively, to qualify for that designation. Using this characterization, we were able to determine whether there is a relationship between active learning and 3D learning using a chi-square analysis. This type of analysis can determine whether co-occurrences of the two are greater than one would expect by chance. Out of the 417 video segments in our data set, more than half of them are neither 3D nor active. This is not surprising, as traditional instruction is common, particularly in large-enrollment introductory courses that make up the majority of our data set, and studies have shown that the transformation of instruction is slow [9]. 93 of the 417 segments were characterized as 3D, while 153 were active. 58 of these active sections were also 3D and the rest (95) were not 3D. Using these data, we find a significant association between 3D and active segments with a small-to-medium effect size (chi-square, 33.9, p_fisher <0.001, phi = 0.285), as shown in Fig 3.

This chi-square test tells us that 3D and active instructional segments tend to co-occur, but it does not tell us what is driving this relationship. A post-hoc analysis using the contributions to the chi-square allowed us to calculate the standardized residual for each of the cells in the contingency table (Fig 3). This provides a measure for how different the observed count is from the expected count, and thus shows which combinations are driving the associations in the table. For this table, the critical value is 9.14, and therefore, as shown in Fig 3, the major driver of significance is the higher-than-expected number of segments in which 3D & active learning co-occur.

Our findings indicate that 3D instruction is moderately associated with active learning, but that the reverse is not true. In our sample, we observed that about 1/3 of the 3D segments were not active (35/93), whereas nearly 2/3 of the active learning segments were not 3D (95/153). That is, a focus on 3DL is also likely to involve active learning, whereas a focus on active learning practices is not as likely to incorporate 3DL.

## Discussion and implications

Our analysis of the video recordings from 82 classes (and 417 total instructional segments), using the 3D-LOP shows that: 1) 3D instruction and active learning are different, and it is possible to have one without the other, and 2) 3D instruction is likely to include active student engagement, whereas active learning does not necessarily include 3DL. Although we do not know how this association arises, there are several potential explanations. For example, it may

be that because the Scientific and Engineering Practices are inherently active, when instructors focus on 3DL and incorporating these practices into their instruction and assessment, they are also more likely to actively engage their students. Alternatively, perhaps instructors who become aware of 3DL strategies are already cognizant of the advantages of active learning and thus incorporate them along with 3DL. It is also possible that because engaging students in Scientific Practices is at the heart of 3DL, instructors who understand 3DL and are intentional in their instructional design also incorporate active learning to engage the students. The mechanism by which an instructor comes to use 3DL almost certainly depends on the instructor and the constraints and affordances of their environment, but what seems clear is that while 3DL and active learning could both be considered evidence-based approaches to teaching and learning, only one of them provides a well described framework and mechanism to support students' use of knowledge in scientifically authentic ways.

We will illustrate this difference using the topic of meiosis in an introductory biology course as an example, and as shown in Table 1. A traditional lecture on meiosis may involve discussing the structures seen in diagrams or micrographs illustrating the stages of meiosis and describing the importance of meiosis in generating gametes. Having students work in groups making diagrams or manipulating pipe cleaners or yarn to illustrate the stages of meiosis converts this lecture into an active learning experience that is not necessarily three-dimensional, as illustrated in Fig 2A. While this type of activity may enhance student performance on assessments that ask them to match images to the stages of meiosis or describe what happens at each stage, it is unclear, or even unlikely, that this engaged learning would enhance their ability to explain how and why meiosis produces gametes with one allele of each gene (assuming a

**Table 1. Distinguishing features of four hypothetical biology class meetings.**

| Biology Example | Not Active (Instructor-centered) | Active (Student-centered) |
|---|---|---|
| Not 3D | • traditional lecture<br>• instructor discusses structures associated with stages of meiosis<br>• instructor describes importance of meiosis in generating gametes<br>• call-and-response questioning or the instructor addresses student questions during the lecture | • multiple tasks that engage students<br>• students identify the correct number of chromosomes for a given set of cells<br>• students draw cells in different stages of cell division<br>• students manipulate pipe cleaners or yarn to illustrate stages<br>• students identify the genotype and phenotype of common traits |
| | Instructor presents content without SPs or CIs or CCCs (content agnostic; didactic) | Students re-represent and identify (content agnostic; interactive) |
| 3D | • traditional lecture<br>• **instructor constructs models** that explain how the molecular machinery in the cell replicates and sorts information<br>• **instructor makes predictions** of what happens to that information when the mechanism fails or changes | • multiple tasks that engage students<br>• **students construct models** and use them to explain how the molecular machinery in the cell replicates and sorts information<br>• **students make predictions** of what happens to that information when the mechanism fails or changes using the model |
| | Instructor demonstrates how to explore a phenomenon with 3DL (CI-focused content with SPs & CCCs; didactic) | Students explore a phenomenon with 3DL (CI-focused content with SPs & CCCs; interactive) |

Distinguishing features of four hypothetical biology class meetings focused on the topic of meiosis, representing each of the four possible combinations of Active and 3D. The class depicted by the timeline in Fig 2A falls into the Active / Not 3D cell here.

diploid organism) while contributing to genetic diversity; or to predict how changes in the components or process would change the outcome, without explicitly asking students to use the model to explain or predict. Helping students achieve these goals requires engagement in constructing and using models to explain how the molecular machinery in the cell replicates and sorts information and how this causes appropriate mixing and distribution of this information. Making a lesson on meiosis three-dimensional accomplishes this. The instructor would focus on the core idea of information flow and include a scientific practice, for example by using a model to make predictions, using the cross-cutting concept cause and effect. The purpose of the lesson is now helping students develop a mechanistic understanding of how the cell sorts genetic information, which enables predictions of what happens to that information when the mechanism fails or changes. As we have shown earlier, engaging students in making predictions will most likely result in active learning, but what the students are doing and why they are doing it will be quite different than in the original active learning scenario. It is certainly possible that active learning can result in high-quality instructional activities that promote knowledge in use or activities that mirror authentic scientific activities, however, three-dimensional learning, by design, intentionally accomplishes this, and in many cases results in active learning as well.

The key difference between engaging students in active learning and engaging students in 3DL is that 3DL requires students to gain experience with the components of the Scientific Practices. For many students, this requires a shift–from restating ideas that they have learned, or learning skills that are never used to predict or explain, or performing calculations without understanding what the result implies–to understanding and articulating why a phenomenon occurs, or using data or calculations to support a claim, or constructing and using models to predict and explain what happens when the system is changed. 3DL provides both instructors and students with explicit guidelines for what it means to engage in Scientific Practices. Rather than the nebulous goals of "critical thinking" or "higher order thinking", 3DL makes it possible to construct both formative and summative assessments that require engagement in 3DL in all its forms. There are a number of resources available for faculty who are interested in adopting the 3DL approach, from the original National Academies report [22], to how to adapt assessments to make them 3D [50], the design of 3D curricula [30,51,52], and approaches to faculty development [53,54]. As noted we have also developed protocols that can be used both to measure change in assessments and instruction, and to support faculty development by highlighting important pedagogical changes and instructional designs that tend to support 3DL, including courses that go beyond the introductory level (since the underlying theory behind the use of 3DL can be (and has been) applied to the design of course sequences and learning tasks). Finally, we acknowledge that there are facts, skills and procedures that students have to learn in order to use them in the context of Scientific Practices. In general, we have found that around 50% of both course time and assessment tasks are an accessible goal that appears to lead to improved and long-term use of knowledge [29].

Because most faculty development is focused on incorporating student-centered pedagogies, such as those discussed in Freeman et. al. [3], this presents us with a dilemma. Should faculty development focus on active engagement, as is currently the case, or should it focus on 3DL, assuming that student engagement will follow? We caution that characterizing transformation efforts by focusing solely on active learning, without also investigating what is expected of students in terms of sensemaking and reasoning may result in the *illusion of transformation*, or what Wiggins and McTighe describe as "*hands on without being minds on*" [55] while at the same time maintaining the status quo.

We propose that it is time to move beyond "active learning" and refocus attention on what it is that we want students to know and do. There are many potential ways to approach

redesigning teaching and learning, for example: a focus on modelling phenomena [56,57], or systems thinking [58–60], the well-established POGIL learning cycle [17], Course based research experiences (CURES) [61], or the use of interactive simulations to model predict and explain data arising from a specific phenomenon [19]. However, 3DL is a unifying framework that can tie all of these approached together. POGIL and the use of interactive simulations to model and predict can be understood as incorporating the Scientific Practices of constructing and using models and analyzing and interpreting data., and Systems Thinking is a cross-cutting concept. It is also important to recognize that none of these approaches are meaningful without connections to disciplinary Core Ideas. 3DL provides an evidence-based framework for designing instructional materials that center around student sensemaking of phenomena, and, as we have shown, tends to include student engagement, even in large lecture sections that traditionally are far more passive. We propose that focusing on creating 3DL aligned learning objectives, assessments, and instructional materials is a more fruitful approach to transforming a learning environment than the more common approach which focuses on instructional methods such as incorporating clickers, or group activities alone. This approach recognizes that it is **what** students are doing, across a wide range of activities both inside and outside of the class, rather than the mere fact that they are doing something, that ultimately can result in students having more coherent, connected, and useful disciplinary knowledge.

## Supporting information

**S1 File.**
(PDF)

**S1 Data.**
(XLSX)

## Acknowledgments

We thank the faculty who provided observational access, engaged in discussions of core ideas, and have taken steps to incorporate three-dimensional learning into their courses. We also thank our advisory board members for their helpful feedback: Stacey Lowery Bretz, Charles Henderson, Joseph Krajcik, Michelle Smith, and Marilyne Stains. Finally, our team of undergraduate research assistants (S. Luba, S. Ly, C. Morrison, K. Noyes, Z. Nusbaum, R.Thornburg, and Q. Babcock) helped code the videos for teaching activities and create 3D-LOP timelines.

## Author Contributions

**Conceptualization:** Melanie M. Cooper, Marcos D. Caballero, Diane Ebert-May, Cori L. Fata-Hartley, Deborah G. Herrington, Lynmarie A. Posey, Sonia M. Underwood.

**Data curation:** Melanie M. Cooper, Cori L. Fata-Hartley, James T. Laverty, Paul C. Nelson.

**Formal analysis:** Melanie M. Cooper, Marcos D. Caballero, Justin H. Carmel, Erin M. Duffy, Diane Ebert-May, Cori L. Fata-Hartley, Deborah G. Herrington, James T. Laverty, Paul C. Nelson, Lynmarie A. Posey, Jon R. Stoltzfus, Ryan L. Stowe, Ryan D. Sweeder, Stuart Tessmer, Sonia M. Underwood.

**Funding acquisition:** Melanie M. Cooper, Marcos D. Caballero, Justin H. Carmel, Diane Ebert-May, Cori L. Fata-Hartley, Deborah G. Herrington, James T. Laverty, Lynmarie A. Posey, Ryan D. Sweeder, Sonia M. Underwood.

**Investigation:** Melanie M. Cooper, Diane Ebert-May, Paul C. Nelson.

**Methodology:** Melanie M. Cooper, Marcos D. Caballero, Deborah G. Herrington, Paul C. Nelson, Lynmarie A. Posey, Jon R. Stoltzfus.

**Project administration:** Melanie M. Cooper, Deborah G. Herrington, Paul C. Nelson.

**Resources:** Melanie M. Cooper.

**Supervision:** Melanie M. Cooper.

**Validation:** Melanie M. Cooper, Marcos D. Caballero, Justin H. Carmel, Erin M. Duffy, Diane Ebert-May, Paul C. Nelson, Lynmarie A. Posey, Jon R. Stoltzfus, Ryan L. Stowe, Ryan D. Sweeder, Stuart Tessmer, Sonia M. Underwood.

**Visualization:** Paul C. Nelson, Ryan L. Stowe.

**Writing – original draft:** Melanie M. Cooper.

**Writing – review & editing:** Melanie M. Cooper, Marcos D. Caballero, Justin H. Carmel, Erin M. Duffy, Diane Ebert-May, Cori L. Fata-Hartley, Deborah G. Herrington, James T. Laverty, Paul C. Nelson, Lynmarie A. Posey, Jon R. Stoltzfus, Ryan L. Stowe, Ryan D. Sweeder, Stuart Tessmer, Sonia M. Underwood.

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
