## [Decision Letter · Decision Letter 0]

11 Jan 2024

PONE-D-23-39689Beyond Active Learning: Using 3-Dimensional Learning to Create Scientifically Authentic, Student-Centered ClassroomsPLOS ONE

Dear Dr. Cooper,

Thank you for submitting your manuscript to PLOS ONE. After careful consideration, we feel that it has merit but does not fully meet PLOS ONE’s publication criteria as it currently stands. Therefore, we invite you to submit a revised version of the manuscript that addresses the points raised during the review process.

It is requested to please submit the supplementary data associated with the manuscript. 

We look forward to receiving your revised manuscript.

Kind regards,

Shailender Kumar Verma, Ph.D.

Academic Editor

PLOS ONE

Journal Requirements:

2. Thank you for stating the following financial disclosure: "National Science Foundation [EHR DUE 1725520]"

3. Please expand the acronym “EHR DUE” (as indicated in your financial disclosure) so that it states the name of your funders in full.

Reviewers' comments:

Reviewer's Responses to Questions

**Comments to the Author**

1. Is the manuscript technically sound, and do the data support the conclusions?

Reviewer #1: No

Reviewer #2: Yes

Reviewer #3: Yes

Reviewer #4: Partly

Reviewer #5: Partly

Reviewer #6: Yes

Reviewer #7: Yes

Reviewer #8: Yes

2. Has the statistical analysis been performed appropriately and rigorously? 

Reviewer #1: I Don't Know

Reviewer #2: Yes

Reviewer #3: Yes

Reviewer #4: No

Reviewer #5: I Don't Know

Reviewer #6: Yes

Reviewer #7: Yes

Reviewer #8: Yes

3. Have the authors made all data underlying the findings in their manuscript fully available?

Reviewer #1: No

Reviewer #2: Yes

Reviewer #3: Yes

Reviewer #4: Yes

Reviewer #5: No

Reviewer #6: Yes

Reviewer #7: Yes

Reviewer #8: Yes

4. Is the manuscript presented in an intelligible fashion and written in standard English?

Reviewer #1: Yes

Reviewer #2: Yes

Reviewer #3: Yes

Reviewer #4: Yes

Reviewer #5: Yes

Reviewer #6: Yes

Reviewer #7: Yes

Reviewer #8: Yes

5. Review Comments to the Author

Reviewer #1: Was there a supporting information file that was not included? The authors provide none of their data and the paper only contains "representative examples" of timelines. Perhaps the submission was incomplete for some technical reason?

As it stands it is impossible to review this submission based on the PLOS criteria.

Reviewer #2: This is an excellent manuscript and an exciting suggestion of how to move beyond active learning. My biggest suggestion is that I was left wanting a little more by way of the discussion/conclusion. I loved figure 1 but I would suggest an additional figure or table later on showing the 3DL areas and how they were specifically applied in this study. Also more discussion of ways to implement etc. I found that you all showed that this is potentially beneficial but a bit more discussion of implication s would be beneficial.

Reviewer #3: As someone who as been following the 3D learning literature over the past couple of years, this paper is greatly appreciated and in my opnion it adds value to the existing body of research. I feel this paper should be accepted for publication, however I do have a few minor suggestions. I hope the authors find these useful and that they might help further improve the manuscript.

1) The authors report that 93 of the 417 instructional segments were identified as 3DL. I would be interested to know how many different instructors taught these 93 segments. If this can be reported, I would then be interested to see a summary of how many of these instructors were exposed to the 3DL literature and how many (if any) may have incorporated 3DL components into their course without explicitly knowing about the 3DL reform movement. In my experience discussing 3DL with colleagues in my department, many of them incorporate aspects of 3DL in their teaching without having ever been exposed to this framework prior to our discussions. If all of the 3DL segments in this study were from faculty who went through some specific 3DL training or were known to have been exposed to the 3DL literature, the authors might note in the discussion that many STEM faculty in higher ed likely value 3DL principles without necessarily being aware of 3DL reform efforts. I think this might help faculty instructors who are not immersed in the chemistry education literature (e.g., research faculty at R1 institutions), but who are doing good work in the classroom feel more valued for their efforts to push their students to engage in meaningful ("3D-esque") learning.

2) In the conlcuding paragraph in the Results/Implications the authors might note that incorporating 3DL in higher ed might need to be thought of in a more holistic way with respect to the four-year curriculum? For instance, though we should obviously aim to incoproate 3DL in our intro general chemistry/bio/physics courses, one could argue there are many skills that students need to learn in these courses which are then used throughout the subsquent curriculum and the prevelance of 3DL in these courses may not be as high. Perhaps there could/should be a continuum of 3DL that increases as students progress through the four-year degree program? I don't think this would change the overall argument being made in this concluding paragraph, but it might provide some valuable nuance.

3) The authors state on pg. 19 that "...only one of them [3DL] provides a mechanism to support students’ use of knowledge in scientifically authentic ways." While I generally agree with this statement, I think some clarification from the authors might be prudent. For instance, if I employed an activity in my class that required students to apply their knowledge of time-integrated rate laws to evaluate concentration and time data to determine/evaluate the stability of two different drug molecules (SPs = Analyzing and Interpreting Data, Engaging in Argument from Evidence, and Constructing Explanations), but I didn't explicitly connect this to the 3DL core ideas or integrate an engineering pratice, would this be 3DL? If not, would this not still be using knowledge in a scientifically authentic way (I would argue yes it is)? The authors might explicitly state what they would consider to be 3DL and not 3DL, and if they were to use a more stringent defintion, they might acknowledge that active learning without 3DL could be in some sort of in-between state in which meaningful/authentic learning is still taking place but a full-fledged 3DL reform has not been carried out.

Reviewer #4: The manuscript explores the concept of three-dimensional learning (3DL) in STEM education, challenging traditional active learning methods by integrating three key components: scientific and engineering practices, disciplinary core ideas, and crosscutting concepts. This approach is inspired by the National Academy's 'A Framework for K-12 Science Education.' The authors have collected data from various courses to assess 3DL's impact in higher education, particularly its role in enhancing student engagement and learning effectiveness. They argue that 3DL promotes not only student engagement but also meaningful interaction with core disciplinary ideas, an aspect often overlooked in traditional active learning.

However, the manuscript could benefit from additional details in several areas. I recommend acceptance after addressing the following points:

1. Implementation of 3DL: More specific examples or case studies demonstrating 3DL's application in diverse educational settings would clarify its implementation.

2. Comparative Analysis: An in-depth comparative analysis with other educational models, especially active learning methods, would enrich the discussion, highlighting 3DL's unique advantages and challenges.

3. Quantitative Evidence: The manuscript would benefit from a broader presentation of quantitative data, such as student performance metrics or statistical analyses, to substantiate the effectiveness of 3DL.

4. Long-term Impact: Discussing the long-term effects of 3DL on students' learning outcomes and career trajectories would provide a more comprehensive view of its benefits.

Reviewer #5: The three strands of “Three-Dimensional Learning (3DL)” are Scientific Practices, Core Ideas, and Crosscutting Concepts. 3DL is a useful framework for student learning in STEM classrooms and, as argued by the authors, is more precisely defined than “active learning.” This paper addresses two questions: (i) what does a class look like when it is engaged in 3DL, active learning, or combinations of the two? And (ii) how are 3DL and active learning related to one another?

The paper summarizes the analysis of 417 video-recorded segments of 82 class meeting sessions in the introductory courses in biology, chemistry, and physics. Diagrams of each of four different instructor pedagogical approaches are shown: active but not 3D, active and 3D, 3D but not active, and neither active nor 3D. Analysis indicates that 3D instruction is moderately associated with active learning, but not the reverse.

I have three comments:

1. While it appears that the authors believe that there is an obvious difference between active learning vs. 3DL, it is not clear how the differences manifest in the practices/behaviors in the classroom. Therefore, for the non-specialist reader, it could be very helpful to illustrate the difference(s). To illustrate the differences as they are operationalized, it would be informative to show how the same material can be presented in two different ways, rather than the present version which reports observations on four different topics and fields. For example, in the first segment (meiosis) diagramed, what could the instructor change so that the class would be an example of 3D or 3D+active?

2. The paper appears to imply that a course that uses 3D is a better way to engage the students than a course that only uses active learning. If my interpretation is correct, then it would be helpful to clearly explain (a chart or table?) the differences and their benefits and shortcomings.

3. In several places, the Introduction appears to dismiss or diminish the significance/efficacy of active learning. If this is the intended message, then I think it can be said once, directly, and with more details (see comment 2, above).

Reviewer #6: Please add more references linked with the qualitative interpretation in your article.

Reviewer #7: The authors address a relevant topic and an interesting research. Although the paper has merit, I would advise some revisions before the paper is published. My major concern has do to with the methodology. First, the authors make an argument on how 'active learning' is ill-defined, but they do not themselves provide a definition and use the same terminology they critique to code classrooms. I would suggest a clarification of this point, which is central, in order to sustain the work on a sound framework. From my perspective, active learning is more than using 'tasks'. If the authors want to prove that 'active learning' should not be the focus in higher education, they have to be clear on what they understand active learning is and is not.

Also, the authors argue that higher-order thinking is also ill-defined by quoting a study with 38 teachers in Kindergarten to Grade 9 classrooms. Given the differences between these school-levels and higher education, I think that this point needs further evidence.

I would also like to have seen a more detailed description of the methodology: when were the video recordings made (the year they started) and were the 4 years in a row or interleaved? Was the inter rater reliability assessed? Why use segments and not the entire class? How were segments defined (give examples of how 'particular topic' can be identified by coders; given that this is your ‘unit’ of observation, we need to be given information on how they can be identified)? Was there a minimum time established for each segment? What was the interval of time between the bigger segment and the smallest one? Time can also be an important variable when active learning or 3D in put into place.

Finally, I do not think that the title of paper is clear. The paper does not provide evidence that active learning does not promote student-centered classrooms or that it is not scientifically authentic, as the title may us lead to believe.

Correct the typo in line 253.

Reviewer #8: Thank you for this fine presentation of 3DLearning.

Students who are able to learn STEM concepts, effectively communicate their newly acquired knowledge either in a group setting or individually and then succesfully apply what has been learned to a new situation or problem engenders excitement in the steps toward mastering a subject. Such STEM learning at the high school level or in the early college years would decrease the number of students with low grades or course withdrawal because learning becomes fun and with purpose. Passion for scientific inquiry in chemistry, biology or physics will be iginited if students understand the why and the how to apply their newly gained knowledge in the world in which they find themselves. Thus, the connection between a student's enjoyment of learning a STEM subject and 3DLearning might be of interest to those reading this manuscript. There is no substitute for a professor or teacher who loves their subject, and can effectively impart that love to his or her students.

6. PLOS authors have the option to publish the peer review history of their article (what does this mean?). If published, this will include your full peer review and any attached files.

Reviewer #1: No

Reviewer #2: No

Reviewer #3: No

Reviewer #4: No

Reviewer #5: No

Reviewer #6: No

Reviewer #7: No

Reviewer #8: **Yes: **Douglas J. Grider, MD

Vice Chair, Basic Science Education

Virginia Tech Carilion School of Medicine

---

## [Author Response · Author response to Decision Letter 0]

29 Feb 2024

Responses to Reviewer Comments (in italics)

Reviewer #1:

Was there a supporting information file that was not included? The authors provide none of their data and the paper only contains "representative examples" of timelines. Perhaps the submission was incomplete for some technical reason?

As it stands it is impossible to review this submission based on the PLOS criteria.

Response: We have uploaded the de-identified coding sheets, which contain the data used in this paper. However, since the original videos contain identifiable faculty we are not able to supply links to those since our IRB is predicated on individual faculty being de-identified. More timelines and data analysis are also available in our paper describing the development of the 3D-LOP, along with supplementary information files that describe the explicit coding procedure. 

Reviewer #2: 

This is an excellent manuscript and an exciting suggestion of how to move beyond active learning. My biggest suggestion is that I was left wanting a little more by way of the discussion/conclusion. I loved figure 1 but I would suggest an additional figure or table later on showing the 3DL areas and how they were specifically applied in this study. Also more discussion of ways to implement etc. I found that you all showed that this is potentially beneficial but a bit more discussion of implication s would be beneficial.

Response: Thank you for this suggestion. We have added a new section highlighting papers that address “How to do 3DL” in different ways, including how to adapt assessments, examples of 3D curricular materials, a description of the Fellowship, and a report of how faculty were impacted. In addition, we have added a section to the discussion specifically discussing how the class that we coded as active but not 3D might have been redesigned to align with the 3DL Framework. 

Reviewer #3: 

As someone who as been following the 3D learning literature over the past couple of years, this paper is greatly appreciated and in my opnion it adds value to the existing body of research. I feel this paper should be accepted for publication, however I do have a few minor suggestions. I hope the authors find these useful and that they might help further improve the manuscript.

1) The authors report that 93 of the 417 instructional segments were identified as 3DL. I would be interested to know how many different instructors taught these 93 segments. If this can be reported, I would then be interested to see a summary of how many of these instructors were exposed to the 3DL literature and how many (if any) may have incorporated 3DL components into their course without explicitly knowing about the 3DL reform movement. In my experience discussing 3DL with colleagues in my department, many of them incorporate aspects of 3DL in their teaching without having ever been exposed to this framework prior to our discussions. If all of the 3DL segments in this study were from faculty who went through some specific 3DL training or were known to have been exposed to the 3DL literature, the authors might note in the discussion that many STEM faculty in higher ed likely value 3DL principles without necessarily being aware of 3DL reform efforts. I think this might help faculty instructors who are not immersed in the chemistry education literature (e.g., research faculty at R1 institutions), but who are doing good work in the classroom feel more valued for their efforts to push their students to engage in meaningful ("3D-esque") learning.

Response: We agree that knowing how many of the 3D classes and segments were the result of specific exposure to 3DL would be interesting, but in practice counting the number of Fellows (and proportion of those 93 segments) is really all we can do. However, there are many ways that faculty may have been exposed to 3DL through influences outside the time and space of the Fellowship (for example, faculty who team teach in a multi -section transformed course alongside alumni of the Fellowship). More broadly, we note that the research question implicated in this comment is beyond the scope of this paper, both because of the difficulties associated with establishing (in)direct exposure to 3DL, and because our IRB commitment to anonymity.

2) In the concluding paragraph in the Results/Implications the authors might note that incorporating 3DL in higher ed might need to be thought of in a more holistic way with respect to the four-year curriculum? For instance, though we should obviously aim to incoproate 3DL in our intro general chemistry/bio/physics courses, one could argue there are many skills that students need to learn in these courses which are then used throughout the subsquent curriculum and the prevelance of 3DL in these courses may not be as high. Perhaps there could/should be a continuum of 3DL that increases as students progress through the four-year degree program? I don't think this would change the overall argument being made in this concluding paragraph, but it might provide some valuable nuance.

Response: We completely agree with the Reviewer that 3DL should not be confined to introductory courses, and should continue throughout the curriculum, and have attempted to clarify our beliefs. It was not our intention to imply that 3DL only applies to gateway courses, but most of the data come from such courses and they were also the initial focus of the MSU fellowship program.

3) The authors state on pg. 19 that "...only one of them [3DL] provides a mechanism to support students’ use of knowledge in scientifically authentic ways." While I generally agree with this statement, I think some clarification from the authors might be prudent. For instance, if I employed an activity in my class that required students to apply their knowledge of time-integrated rate laws to evaluate concentration and time data to determine/evaluate the stability of two different drug molecules (SPs = Analyzing and Interpreting Data, Engaging in Argument from Evidence, and Constructing Explanations), but I didn't explicitly connect this to the 3DL core ideas or integrate an engineering pratice, would this be 3DL? If not, would this not still be using knowledge in a scientifically authentic way (I would argue yes it is)? The authors might explicitly state what they would consider to be 3DL and not 3DL, and if they were to use a more stringent defintion, they might acknowledge that active learning without 3DL could be in some sort of in-between state in which meaningful/authentic learning is still taking place but a full-fledged 3DL reform has not been carried out.

Response: This is a good point, and in our work we have consistently stated that not every class activity is (or should be) 3D (e.g., Matz 2018), and have reiterated that here. The activity described in the Reviewer’s comment is clearly important and useful, and certainly engages students in scientific practices. The point we have tried to make is that as students develop disciplinary expertise, at some point the skills and knowledge they gain must be connected to core ideas (not necessarily in every class in every activity). The goal is to avoid a course where techniques are solely emphasized in the absence of disciplinary ideas. 3DL provides a concrete path to authentic engagement and (the current study says) is also more likely to actively engage students. 

Reviewer #4: 

The manuscript explores the concept of three-dimensional learning (3DL) in STEM education, challenging traditional active learning methods by integrating three key components: scientific and engineering practices, disciplinary core ideas, and crosscutting concepts. This approach is inspired by the National Academy's 'A Framework for K-12 Science Education.' The authors have collected data from various courses to assess 3DL's impact in higher education, particularly its role in enhancing student engagement and learning effectiveness. They argue that 3DL promotes not only student engagement but also meaningful interaction with core disciplinary ideas, an aspect often overlooked in traditional active learning.

However, the manuscript could benefit from additional details in several areas. I recommend acceptance after addressing the following points:

1. Implementation of 3DL: More specific examples or case studies demonstrating 3DL's application in diverse educational settings would clarify its implementation.

2. Comparative Analysis: An in-depth comparative analysis with other educational models, especially active learning methods, would enrich the discussion, highlighting 3DL's unique advantages and challenges.

3. Quantitative Evidence: The manuscript would benefit from a broader presentation of quantitative data, such as student performance metrics or statistical analyses, to substantiate the effectiveness of 3DL.

4. Long-term Impact: Discussing the long-term effects of 3DL on students' learning outcomes and career trajectories would provide a more comprehensive view of its benefits.

Response: While some of these suggestions are certainly fruit for further studies, we believe that they are all beyond the scope of the current paper. The paper is not intended to be a primer on 3DL - especially since we have published earlier studies on these topic.:

Suggestion #1. We have added a short section to point readers to our published papers that help clarify implementation. Case studies of active and 3DL classes would be an interesting study, but are beyond the scope of this paper. 

Suggestion #2. We have added a section to the discussion specifically addressing how the class that we coded as active but not 3D might have been redesigned to align with the 3DL Framework.

Suggestion #3. We have published extensively on comparisons of 3DL courses and more traditional (be they “active” or “traditional lecture”) and have discussed several of them including r citations 20-31.

Suggestion #4. It is very difficult to track students as they move through multiple curricula and majors at such a large university. We certainly do not have the resources to follow students into their career trajectories. We do have data on students as they move through 4 semesters of chemistry, and we have added a short section indicating that students from the large enrollment 3D general chemistry courses do as well or better than students from more selective (honors and majors) courses in the subsequent organic chemistry courses.

Reviewer #5: 

The three strands of “Three-Dimensional Learning (3DL)” are Scientific Practices, Core Ideas, and Crosscutting Concepts. 3DL is a useful framework for student learning in STEM classrooms and, as argued by the authors, is more precisely defined than “active learning.” This paper addresses two questions: (i) what does a class look like when it is engaged in 3DL, active learning, or combinations of the two? And (ii) how are 3DL and active learning related to one another?

The paper summarizes the analysis of 417 video-recorded segments of 82 class meeting sessions in the introductory courses in biology, chemistry, and physics. Diagrams of each of four different instructor pedagogical approaches are shown: active but not 3D, active and 3D, 3D but not active, and neither active nor 3D. Analysis indicates that 3D instruction is moderately associated with active learning, but not the reverse.

I have three comments:

1. While it appears that the authors believe that there is an obvious difference between active learning vs. 3DL, it is not clear how the differences manifest in the practices/behaviors in the classroom. Therefore, for the non-specialist reader, it could be very helpful to illustrate the difference(s). To illustrate the differences as they are operationalized, it would be informative to show how the same material can be presented in two different ways, rather than the present version which reports observations on four different topics and fields. For example, in the first segment (meiosis) diagramed, what could the instructor change so that the class would be an example of 3D or 3D+active?

2. The paper appears to imply that a course that uses 3D is a better way to engage the students than a course that only uses active learning. If my interpretation is correct, then it would be helpful to clearly explain (a chart or table?) the differences and their benefits and shortcomings.

Response (to both comments 1 & 2): We thank the reviewer for this useful suggestion. A paragraph and associated table has been added to the revised discussion to address this comment (and Reviewer #2’s similar suggestions). By using the example “active but not 3D” class meeting that was presented in one of the timelines as an anchor, we illustrate the type of instructional re-framing that is required to provide students opportunities to engage with 3DL. We believe these comments have led to greater clarity in the revised manuscript in our effort to highlight that such a transformation is not a shift or tweak in an instructional strategy, but a rethinking of what is valued as important that students know and do.

3. In several places, the Introduction appears to dismiss or diminish the significance/efficacy of active learning. If this is the intended message, then I think it can be said once, directly, and with more details (see comment 2, above).

Response: Our intent was not to diminish the work on active learning that has gone before, but rather to point out (as has been noted by others) that it is ill defined, and simply adding an student engagement component does not ensure deeper or more authentic learning. We hope the addition of the table and following discussion makes this clearer.

Reviewer #6: 

Please add more references linked with the qualitative interpretation in your article.

Response: We have added more references and more discussion in the implications section as discussed earlier.

Reviewer #7: 

The authors address a relevant topic and an interesting research. Although the paper has merit, I would advise some revisions before the paper is published. My major concern has do to with the methodology. First, the authors make an argument on how 'active learning' is ill-defined, but they do not themselves provide a definition and use the same terminology they critique to code classrooms. I would suggest a clarification of this point, which is central, in order to sustain the work on a sound framework. From my perspective, active learning is more than using 'tasks'. If the authors want to prove that 'active learning' should not be the focus in higher education, they have to be clear on what they understand active learning is and is not.

Response: We are not the first or the only people to who have pointed out that active learning may be in the “eye of the beholder” and means different things to different people. As we state in the paper we have used the Theobald definition(s) for student engagement (reference 5), and in our earlier paper on the development of the 3D-LOP we defined the class activities that could engage students, for example tasks, or clicker questions. We also coded whether these activities were instructor led or involved student engagement. We have also added more discussion around Table 1 and the meiosis example to make the point that we do want active learning, we just want it to be purposeful activity and 3DL provides purpose.

Also, the authors argue that higher-order thinking is also ill-defined by quoting a study with 38 teachers in Kindergarten to Grade 9 classrooms. Given the differences between these school-levels and higher education, I think that this point needs further evidence.

Response: We have added more references specifically focussed on higher education and provided examples of other researchers who support this viewpoint.

I would also like to have seen a more detailed description of the methodology: when were the video recordings made (the year they started) and were the 4 years in a row or interleaved? Was the inter rater reliability assessed? Why use segments and not the entire class? How were segments defined (give examples of how 'particular topic' can be identified by coders; given that this is your ‘unit’ of observation, we need to be given information on how they can be identified)? Was there a minimum ti

---

## [Decision Letter · Decision Letter 1]

21 Mar 2024

PONE-D-23-39689R1Beyond Active Learning: Using 3-Dimensional Learning to Create Scientifically Authentic, Student-Centered ClassroomsPLOS ONE

Dear Dr. Cooper,

Thank you for submitting your manuscript to PLOS ONE. After careful consideration, we feel that it has merit but does not fully meet PLOS ONE’s publication criteria as it currently stands. Therefore, we invite you to submit a revised version of the manuscript that addresses the points raised during the review process.

We look forward to receiving your revised manuscript.

Kind regards,

Shailender Kumar Verma, Ph.D.

Academic Editor

PLOS ONE

Reviewers' comments:

Reviewer's Responses to Questions

**Comments to the Author**

1. If the authors have adequately addressed your comments raised in a previous round of review and you feel that this manuscript is now acceptable for publication, you may indicate that here to bypass the “Comments to the Author” section, enter your conflict of interest statement in the “Confidential to Editor” section, and submit your "Accept" recommendation.

Reviewer #1: All comments have been addressed

Reviewer #2: (No Response)

Reviewer #3: (No Response)

Reviewer #4: All comments have been addressed

Reviewer #7: All comments have been addressed

Reviewer #8: All comments have been addressed

2. Is the manuscript technically sound, and do the data support the conclusions?

Reviewer #1: Yes

Reviewer #2: Yes

Reviewer #3: Yes

Reviewer #4: Yes

Reviewer #7: Partly

Reviewer #8: Yes

3. Has the statistical analysis been performed appropriately and rigorously? 

Reviewer #1: Yes

Reviewer #2: Yes

Reviewer #3: Yes

Reviewer #4: No

Reviewer #7: N/A

Reviewer #8: Yes

4. Have the authors made all data underlying the findings in their manuscript fully available?

Reviewer #1: Yes

Reviewer #2: Yes

Reviewer #3: Yes

Reviewer #4: Yes

Reviewer #7: Yes

Reviewer #8: Yes

5. Is the manuscript presented in an intelligible fashion and written in standard English?

Reviewer #1: Yes

Reviewer #2: Yes

Reviewer #3: Yes

Reviewer #4: Yes

Reviewer #7: Yes

Reviewer #8: Yes

6. Review Comments to the Author

Reviewer #1: The authors have addressed my concerns about data availability.

This is now a very useful and technically sound manuscript that offers the following significant advances:

- demonstrates the practical utility and reliability of the 3D-LOP protocol for coding videos of instruction

- shows that active learning and 3DL can be identified separately from videos

- finds that (within their dataset) active learning does not imply 3DL, but typically 3DL does imply active learning

- offers support for instructors who wish to incorporate 3DL into their teaching, and assess their 3DL using the tools described here

Reviewer #2: Thank you for your revisions, I believe you have addressed all of my previous comments as well as many of the comments left by other reviewers. I noticed a few very small typos etc. that you may want to correct before publication: There appears to be a missed space on line 85; a missed period on lines 173, 202, 246, and 477; and typos on lines 456 and 480.

Reviewer #3: PONE-D-23-39689R1

Reviewer Comments:

The authors have addressed most of the concerns and suggestions from the original reviewers, though in many cases in a minimal fashion. Ultimately, I think this paper is worthy of publication in PLOS ONE in its current form, however I feel there are still a couple of significant flaws in the broader arguments the authors are making. If these are not addressed, I suspect many readers will look at this with a more critical view and the reach/impact of this paper will be hindered (especially with respect to impacting classroom practitioners, but also with educational researchers).

1. I feel the authors are conflating the 3DL learning framework with evidence-based instructional practices. It seems to me that 3DL is a framework which acts as a base/guide for building a set of course learning outcomes (particularly through the core ideas and crosscutting concepts), and one could easily argue the scientific/engineering practices are themselves a broader set of additional content-agnostic learning objectives (e.g., developing and using models, analyzing/interpreting data, engaging in argument from evidence, etc. are things I hope my students can do by the end of my course, but they are not distinct classroom practices). Table 1 in the manuscript highlights this point, with the fact that two completely different classroom practices (e.g., “non-active” passive instructor lecture or student-centered active learning) can both lead to 3D learning. I strongly encourage the authors to reconsider highlighting how different types of evidence-based practices can lead to learning outcomes that fit within the framework of 3DL. I have highlighted here three examples of classroom practices and representative publications that I would consider active, AND which aim to achieve the kind of learning outcomes described by the 3DL framework (this is by no means an exhaustive list of evidence-based practices that might promote 3DL).

PhET activities:

a) https://pubs.acs.org/doi/10.1021/ed4005084

b) https://pubs.acs.org/doi/10.1021/acs.jchemed.0c00470

c) https://doi.org/10.1039/D1RP00086A

POGIL:

a) https://pubs.acs.org/doi/10.1021/bk-2008-0994.ch018

b) https://pubs.acs.org/doi/10.1021/acs.jchemed.0c00355

c) https://doi.org/10.1039/C5RP00207A

CURES

a) https://pubs.acs.org/doi/10.1021/acs.jchemed.1c01179

b) https://doi.org/10.1021/acs.jchemed.3c00570

2. I also continue to take issue with two statements made by the authors:

i) “Most studies on the impact of active learning on student outcomes rely either on scores on conceptual multiple choice exams or course grades, but typically little information is provided about what those grades are measuring, and whether they emphasize factual recall or use of knowledge.”

The representative PhET and POGIL articles listed above are examples of studies that use more than course grades to assess the impact of the evidence-based practices on student learning. I feel the authors are overlooking a significant amount of previous research that has been done to assess the impact of evidence-based practices on meaningful student learning, and if they choose to ignore this in the final version of this article, that will likely reduce the persuasiveness of this paper on science educators.

ii) “…but what seems clear is that while 3DL and active learning could both be considered evidence-based approaches to teaching and learning, only one of them provides a well described framework and mechanism to support students’ use of knowledge in scientifically authentic ways.”

The representative articles above were chosen because these studies also provide examples of “using knowledge in scientifically authentic ways.” If the authors were to acknowledge the various types of evidence-based practices that can/do promote 3D learning, they could further amend this statement to highlight how 3DL is a framework of learning that promotes the use of learning objectives aimed at the authentic use of knowledge, which can then be achieved either through explicit “3DL” curricula (e.g,. CLUE) or other evidence-based practices (POGIL, PhET concept development activities, problem-based case studies, CURE lab experiences, etc.). Again, if the authors choose to not to acknowledge these other classroom practices that promote 3DL, it will in my opinion significantly weaken the paper.

Reviewer #4: The authors addressed the review comments thoroughly in the response. No further major and minor revisions are suggested for this article to be considered for full publication.

Reviewer #7: I agree with the authors that there may be a confusion about the definition of active learning, but I do not feel that the discussion may be dismissed just by saying that, especially if you are using a definition of AL to validate your work (in terms of the criteria to code segments). The authors consider AL to just be a number of in-class tasks, but Prince (2004) describes it as “any instructional method that engages students in the learning process. In short, active learning requires students to do meaningful learning activities and think about what they are doing”. How do the authors incorporate the reflexive nature of AL in your model, for instance in the example presented in table 1?

Finally, I have still two minor questions. There was no answer to my question on if the inter rater reliability of coding was assessed. I think this information should be in the methodology. And again, I do not interpreter the title as the authors seem to do and it seems that to me that it goes beyond what is discussed in the article, suggesting that Al is not as scientifically authentic.

Reviewer #8: I wish the connection between the "love" of a subject and how it is taught could be addressed via 3DLearning. I think that professors or teachers who want the best for their students would be flexible in how material is presented and thus assimilated by students.

7. PLOS authors have the option to publish the peer review history of their article (what does this mean?). If published, this will include your full peer review and any attached files.

Reviewer #1: No

Reviewer #2: No

Reviewer #3: No

Reviewer #4: No

Reviewer #7: No

Reviewer #8: No

---

## [Author Response · Author response to Decision Letter 1]

17 Apr 2024

PONE-D-23-39689R1

Beyond Active Learning: Using 3-Dimensional Learning to Create Scientifically Authentic, Student-Centered Classrooms

PLOS ONE

Responses to Reviewers (in italics)

Reviewer #1: The authors have addressed my concerns about data availability.

This is now a very useful and technically sound manuscript that offers the following significant advances:

- demonstrates the practical utility and reliability of the 3D-LOP protocol for coding videos of instruction

- shows that active learning and 3DL can be identified separately from videos

- finds that (within their dataset) active learning does not imply 3DL, but typically 3DL does imply active learning

- offers support for instructors who wish to incorporate 3DL into their teaching, and assess their 3DL using the tools described here

Reviewer #2: Thank you for your revisions, I believe you have addressed all of my previous comments as well as many of the comments left by other reviewers. I noticed a few very small typos etc. that you may want to correct before publication: There appears to be a missed space on line 85; a missed period on lines 173, 202, 246, and 477; and typos on lines 456 and 480.

Response: Thank you for pointing these out, we have fixed them.

Reviewer #3: PONE-D-23-39689R1

Reviewer Comments:

The authors have addressed most of the concerns and suggestions from the original reviewers, though in many cases in a minimal fashion. Ultimately, I think this paper is worthy of publication in PLOS ONE in its current form, however I feel there are still a couple of significant flaws in the broader arguments the authors are making. If these are not addressed, I suspect many readers will look at this with a more critical view and the reach/impact of this paper will be hindered (especially with respect to impacting classroom practitioners, but also with educational researchers).

1. I feel the authors are conflating the 3DL learning framework with evidence-based instructional practices. It seems to me that 3DL is a framework which acts as a base/guide for building a set of course learning outcomes (particularly through the core ideas and crosscutting concepts), 

Response: We agree, that 3DL is a framework that allows us to build course learning outcomes – and just as importantly their aligned assessments. Just as with POGIL or modelling approaches, this is an approach to evidence-based instruction. To remove the confusion about what we mean be evidence based instructional practices we have removed any mention of EBIPS and instead have used the term “active learning” which has emerged as a “catchall” for any kind of student engagement activity. Our finding here is that enacting 3DL is likely to promote student engagement in meaningful scientific practices and will likely result in active learning whereas enacting active learning without careful attention to what is expected of students will certainly engage students but there is no grantee they will be engaged in meaningful activity. This does not, of course, mean that 3DL is the only framework that can engage students in this way, and we have tried to make this point clearer in the manuscript (see below). 

and one could easily argue the scientific/engineering practices are themselves a broader set of additional content-agnostic learning objectives (e.g., developing and using models, analyzing/interpreting data, engaging in argument from evidence, etc. are things I hope my students can do by the end of my course, but they are not distinct classroom practices) Table 1 in the manuscript highlights this point, with the fact that two completely different classroom practices (e.g., “non-active” passive instructor lecture or student-centered active learning) can both lead to 3D learning. 

Response: Here we must respectfully disagree with the reviewer, scientific practices such as developing and using models, analyzing/interpreting data, engaging in argument from evidence, etc. typically have particular knowledge construction goals, and as we note in the manuscript there is ample evidence in the literature that content agnostic learning objectives are unlikely to produce the outcomes desired. The 3DL framework requires that all three dimensions (Core ideas, Scientific Practices and Crosscutting concepts) be tightly integrated. The 3DL framework specifies both the content and what students should do with that content. 

I strongly encourage the authors to reconsider highlighting how different types of evidence-based practices can lead to learning outcomes that fit within the framework of 3DL. I have highlighted here three examples of classroom practices and representative publications that I would consider active, AND which aim to achieve the kind of learning outcomes described by the 3DL framework (this is by no means an exhaustive list of evidence-based practices that might promote 3DL).

PhET activities:

a) https://pubs.acs.org/doi/10.1021/ed4005084

b) https://pubs.acs.org/doi/10.1021/acs.jchemed.0c00470

c) https://doi.org/10.1039/D1RP00086A

POGIL:

a) https://pubs.acs.org/doi/10.1021/bk-2008-0994.ch018

b) https://pubs.acs.org/doi/10.1021/acs.jchemed.0c00355

c) https://doi.org/10.1039/C5RP00207A

CURES

a) https://pubs.acs.org/doi/10.1021/acs.jchemed.1c01179

b) https://doi.org/10.1021/acs.jchemed.3c00570

Response: We agree with the reviewer that explicitly using 3DL as a framework for designing instruction and assessments is not the only framework that might promote 3DL (and indeed one of the PhET activities called out here https://pubs.acs.org/doi/10.1021/acs.jchemed.0c00470, was designed by two of the co-authors on this paper, and guided by their work on the longstanding 3DL4US project part of which is described here). 

We have attempted to clarify that there are other evidence-based approaches that – even though they are not explicitly designed using 3DL – do have the potential to be 3D. For example, POGIL activities typically begin with experimental data that students analyze and interpret (SEP) to make arguments (SEP) about a phenomenon (ideally linked to a core idea and a crosscutting concept). 3DL has the advantage of being more granular than a descriptor like POGIL or CURE. Its utility derives in part from this granularity. It is possible to describe a CURE or POGIL activity that engages students in a scientific practice, crosscutting concept and core idea. 

Although PhET is not an instructional approach – the way the simulations are be used can lend them to analysis and interpretation of data, argumentation and other scientific practice.

Again, our intention was not to imply that there is no effective and impactful prior work, but rather to show that focusing only on student engagement, that is “active learning” without a concomitant emphasis on what we expect students to know and to do, may not lead to desired outcomes.

2. I also continue to take issue with two statements made by the authors:

i) “Most studies on the impact of active learning on student outcomes rely either on scores on conceptual multiple choice exams or course grades, but typically little information is provided about what those grades are measuring, and whether they emphasize factual recall or use of knowledge.”

The representative PhET and POGIL articles listed above are examples of studies that use more than course grades to assess the impact of the evidence-based practices on student learning. I feel the authors are overlooking a significant amount of previous research that has been done to assess the impact of evidence-based practices on meaningful student learning, and if they choose to ignore this in the final version of this article, that will likely reduce the persuasiveness of this paper on science educators.

Response: We have attempted to make our position clearer about the studies that rely on concept tests or course grades (it is the oft cited meta studies such as Freeman et al. – PNAS, 2014, 111(23, 8410-8415). Additionally, a review of research on POGIL has this to say about the studies “We find that this body of research has typically focused on quantitative measures that broadly assess student outcomes, with fewer studies concentrating on qualitative, theory-based explanations of student learning.” (Rodrigues et al. J. Chem Educ. 2020, 97, 3506-3520)

ii) “…but what seems clear is that while 3DL and active learning could both be considered evidence-based approaches to teaching and learning, only one of them provides a well described framework and mechanism to support students’ use of knowledge in scientifically authentic ways.”

The representative articles above were chosen because these studies also provide examples of “using knowledge in scientifically authentic ways.” If the authors were to acknowledge the various types of evidence-based practices that can/do promote 3D learning, they could further amend this statement to highlight how 3DL is a framework of learning that promotes the use of learning objectives aimed at the authentic use of knowledge, which can then be achieved either through explicit “3DL” curricula (e.g,. CLUE) or other evidence-based practices (POGIL, PhET concept development activities, problem-based case studies, CURE lab experiences, etc.). Again, if the authors choose to not to acknowledge these other classroom practices that promote 3DL, it will in my opinion significantly weaken the paper.

Response: We agree with the reviewer and have added statements to this effect in the narrative, emphasizing that the use of “active learning” as a catchall for all methods of student engagement is problematic.

Reviewer #4: The authors addressed the review comments thoroughly in the response. No further major and minor revisions are suggested for this article to be considered for full publication.

Reviewer #7: I agree with the authors that there may be a confusion about the definition of active learning, but I do not feel that the discussion may be dismissed just by saying that, especially if you are using a definition of AL to validate your work (in terms of the criteria to code segments). The authors consider AL to just be a number of in-class tasks, but Prince (2004) describes it as “any instructional method that engages students in the learning process. In short, active learning requires students to do meaningful learning activities and think about what they are doing”. How do the authors incorporate the reflexive nature of AL in your model, for instance in the example presented in table 1?

Response: We used the definition of active learning found in the most cited study (Freeman et al. – PNAS, 2014, 111(23, 8410-8415) on the impact of active learning (that is activities that engage students – discussion, clickers, group activities and so on – just as Prince describes). We use these criteria (which are very similar to the COPUS criteria, https://doi.org/10.1187/cbe.13-08-0154) to characterize what students and faculty are doing in the classroom. Our point is that it matters what students are learning and what they are doing with their knowledge. We hope that this has been made clearer now. We have also emphasized that 3DL is not the only approach to engaging students in authentic scientific practice, and that many transformation efforts may align with 3DL. That is 3DL is an overarching framework that can connect disparate transformation approaches that involve students using knowledge to predict, explain, model and so on.

Finally, I have still two minor questions. There was no answer to my question on if the inter rater reliability of coding was assessed. I think this information should be in the methodology. 

Response: As we note in the manuscript “Each video was coded by at least two disciplinary experts, and if the IRR fell below 70% a third disciplinary expert also coded the videos. Eventually all videos were coded until all the coders agreed.” The full description of the development and validation of the 3D-LAP were discussed in a prior published report, as noted in the manuscript. 

And again, I do not interpreter the title as the authors seem to do and it seems that to me that it goes beyond what is discussed in the article, suggesting that Al is not as scientifically authentic.

Response: Again, as many authors have noted, the term active learning can mean many things – one of our goals is to encourage researchers and practitioners alike to better specify what is meant, which is also one of the findings in the large study on Active Learning (https://journals.sagepub.com/doi/10.1177/1529100620973974). Currently the term is a catchall, that can mean many different things (for example at a recent conference it was pointed out that some faculty thought that showing a video clip in class constituted active learning), and as we have found, activities that fall within this term may not engage students in the use of knowledge in an authentic way 

Reviewer #8: I wish the connection between the "love" of a subject and how it is taught could be addressed via 3DLearning. I think that professors or teachers who want the best for their students would be flexible in how material is presented and thus assimilated by students.

Response: As we noted in the prior set of responses, this is an important topic, but we have no data from our studies that would allow us to make this claim. It is beyond the scope of the report.

---

## [Decision Letter · Decision Letter 2]

23 Apr 2024

Beyond Active Learning: Using 3-Dimensional Learning to Create Scientifically Authentic, Student-Centered Classrooms

PONE-D-23-39689R2

Dear Dr. Cooper,

We’re pleased to inform you that your manuscript has been judged scientifically suitable for publication and will be formally accepted for publication once it meets all outstanding technical requirements.

Kind regards,

Shailender Kumar Verma, Ph.D.

Academic Editor

PLOS ONE

Additional Editor Comments (optional):

All the reviewers (3 -- R1 & 3 -- R2) has unanimously recommended to accept the manuscript with positive feedback. It is suggested to please have look onto few of the minor edits/ typos at proofreading stage (please see, Review Comments to the Author). Thank you for submitting in PLOS ONE and best wishes for your educational research work.

Reviewers' comments:

Reviewer's Responses to Questions

**Comments to the Author**

1. If the authors have adequately addressed your comments raised in a previous round of review and you feel that this manuscript is now acceptable for publication, you may indicate that here to bypass the “Comments to the Author” section, enter your conflict of interest statement in the “Confidential to Editor” section, and submit your "Accept" recommendation.

Reviewer #2: All comments have been addressed

Reviewer #3: All comments have been addressed

Reviewer #7: All comments have been addressed

2. Is the manuscript technically sound, and do the data support the conclusions?

Reviewer #2: Yes

Reviewer #3: Yes

Reviewer #7: Yes

3. Has the statistical analysis been performed appropriately and rigorously? 

Reviewer #2: Yes

Reviewer #3: Yes

Reviewer #7: Yes

4. Have the authors made all data underlying the findings in their manuscript fully available?

Reviewer #2: Yes

Reviewer #3: Yes

Reviewer #7: Yes

5. Is the manuscript presented in an intelligible fashion and written in standard English?

Reviewer #2: Yes

Reviewer #3: Yes

Reviewer #7: Yes

6. Review Comments to the Author

Reviewer #2: Thank you for your edits. My only suggestion is that you pay close attention to small spelling and grammar errors in the final copyediting. I noticed double periods on lines 90 and 530. However these are very minor errors. The content is great and you have done a wonderful job addressing the reviewer comments throughout the revisions.

Reviewer #3: I feel the authors have addressed the most recent comments in a thoughtful manner, and in my opinion this has significantly improved the manuscript. I have just one final minor edit/suggestion the authors may want to address in the final draft of the manuscript prior to publication.

1. For the new passage on page 6, the authors should provide citations for the text callouts to POGIL and CURES.

"There are of several other approaches to transformation of STEM education that, while not explicitly developed using a 3D framework, also have the potential to engage students in the use of scientific practices. For example, POGIL activities often begin with experimental data that students may analyze and interpret, to make an argument about the phenomenon under study. Similarly, Course Based Undergraduate Research Experiences (CURES), may ask students to design and carry out experiments, analyze and interpret data and so on. 3DL is a unifying framework that can tie all these approaches together."

Reviewer #7: Even though I remain skeptical of the definition of active Learning used for this work (I feel that the authors use the one that better fits their 'narrative'), I acknowledge that the work has merit and is worthy of publication.

7. PLOS authors have the option to publish the peer review history of their article (what does this mean?). If published, this will include your full peer review and any attached files.

Reviewer #2: No

Reviewer #3: No

Reviewer #7: No

---

## [Editor Report · Acceptance letter]

29 Apr 2024

PONE-D-23-39689R2 

PLOS ONE

Dear Dr. Cooper, 

I'm pleased to inform you that your manuscript has been deemed suitable for publication in PLOS ONE. Congratulations! Your manuscript is now being handed over to our production team.

Kind regards, 

on behalf of

Dr. Shailender Kumar Verma 

Academic Editor

PLOS ONE